# Micropore Structural Heterogeneity of Siliceous Shale Reservoir of the Longmaxi Formation in the Southern Sichuan Basin, China

**Hu Li [1,2,\*], Hongming Tang [1,2,\*] and Majia Zheng [1,2]**

[1]   State Key Laboratory of Oil and Gas Reservoir Geology and Exploitation, Southwest Petroleum University, Chengdu 610500, China; majiaz_cq@petrochina.com.cn

[2]   School of Geoscience and Technology, Southwest Petroleum University, Chengdu 610500, China

\*   Correspondence: lihu860628@126.com (H.L.); hmtangswpi@126.com (H.T.); Tel.: +86-028-8303-5177 (H.L.); +86-028-8303-7128 (H.T.)

**Abstract:** In recent years, the shale gas in the southern Sichuan Basin has achieved great commercial development, and the Silurian Longmaxi Formation is the main development stratum. In order to solve the problems of great difference production and inaccurate gas content of the Longmaxi Formation shale gas field in the southern Sichuan Basin, based on thin section identification, argon ion polishing-field emission scanning electron microscopy, high pressure mercury injection, low temperature nitrogen adsorption and the fractal method, the micropore structural heterogeneity of the siliceous shale reservoir of the Longmaxi Formation has been studied. The results show the following: The pores of siliceous shale are mainly intergranular pores and organic pores. Image analysis shows that there are obvious differences in size and distribution of shale pores among different types. The micropore structural heterogeneity is as follows: intragranular pore > intergranular pore > organic pore. In the paper, the combination of low temperature nitrogen adsorption method and high-pressure mercury injection method is proposed to characterize the micropore size distribution and fractal dimension, which ensures the credibility of pore heterogeneity. The shale pores are mainly composed of mesopores (2–20 nm), followed by macropores (100–300 nm). For different pore sizes, the fractal dimension from large to small is mesopore, micropore and macropore. Shale pore structure and fractal dimension are correlated with mineral composition and total organic carbon (TOC) content, but the correlation is significantly different in different areas, being mainly controlled by the sedimentary environment and diagenesis.

**Keywords:** pore structure; pore type; heterogeneity; fractal dimension; mineral composition; Longmaxi Formation

---

## 1. Introduction

Marine shale is well developed in China, with huge thicknesses of source rock and favourable conditions for hydrocarbon generation. Among them, the Cambrian Qiongzhusi Formation and the Silurian Longmaxi Formation in the Sichuan Basin have the greatest exploration potential [1–5]. In recent years, following the guidance of the North American, the exploration and development of shale gas in China has progressed remarkably. In 2018, the gas output of the Fuling area and southern Sichuan exceeded $6 \times 10^9$ m$^3$ and $3 \times 10^9$ m$^3$, respectively, which set off an upsurge of shale gas exploration and development in China [6–8].

Although shale gas is the "continuous" gas reservoir, with the evolution of shale gas exploration and development, the heterogeneity and complexity of the shale reservoir has become more and more obvious, and the production difference between interlayer and adjacent wells is obvious,

such as the Barnett shale in the Fort Worth Basin, USA [9]. The daily production of Y203-H2 and Y201-H1 in the Fushun–Yongchuan block is $6 \times 10^9$ m$^3$ and $3 \times 10^9$ m$^3$, respectively, and the reservoir heterogeneity is the most important reason [10]. Previous studies have proved that there are many factors, such as pore structure, organic carbon content, mineral composition, gas content and fracture development affecting the heterogeneity of the shale reservoir, among which pore structure is one of the most important factors [11,12]. As the shale gas reservoir is characterized by sedimentary bedding, organic nanopore development and strong grain orientation, the parameters and methods for evaluating the heterogeneity of unconventional reservoirs such as tight gas sandstone are not fully applicable to shale reservoir [13–20]. In recent years, many qualitative and quantitative multidisciplinary techniques have been used to characterize shale pores. The qualitative techniques include polarized light microscopy, field emission scanning electron microscopy (FE-SEM), and atomic force microscopy [21–23]. The quantitative techniques include low pressure gas adsorption (N$_2$ and CO$_2$), high-pressure MIP, small angle and ultra-small-angle neutron scattering (SANS and USANS), and nano-CT [24–26]. It is generally necessary to comprehensively use a variety of methods to characterize the pore structure due to the complexity of pore structure [27]. Total organic carbon (TOC) is an important index of shale reservoir, which can effectively evaluate the abundance of shale organic matter. At present, TOC content is mainly determined by geochemical experimental analysis method [28].

In addition, because of the strong heterogeneity of the shale pore structure, fractal theory provides a new method for its quantitative study. Fractal geometry studies objects or systems in nature, which are fragmentary, complex and scale-free but self-similar, and can be expressed by the fractal dimension. The fractal dimension (D) is an important parameter used to characterize the heterogeneity and statistical self-similarity of objects and is widely used in Earth science. The fractal dimension can be used to study complex rock pore networks, indicating the degree of heterogeneity of pore structures. In 1983, Pfeifer and Avnir [29] first discovered that pore structure has fractal characteristics when using a molecular adsorption method to study rock pores. In recent years, many studies have been conducted on the fractal characteristics of the pores of sandstone, carbonate and coal. The fractal dimension of 2–3 is consistent within the fractal meaning of the pore system; the closer to 2, the more regular of the pore surface is and the closer to 3, the more complex the pore structure is and the stronger the heterogeneity is. However, there are few preliminary studies that focus solely on shale pores [30,31]. The fractal dimension is used to quantitatively characterize the heterogeneity of the pore structure, which is helpful to understand the gas adsorption and storage capacity of a shale gas reservoir. Therefore, based on the cores of Z101, N210 and W206 wells in the southern Sichuan Basin, this paper evaluated the micropore structural heterogeneity of marine shale and provided guidance for the study area and other shale areas in China.

## 2. Geological Setting

The Sichuan Basin is located in the western Yangtze quasi-platform, with Dalou Mountain in the east, Longmen Mountain fold belt in the west, Daxiangling-Lou Mountain fold belt in the south and Micang Mountain uplift–Daba Mountain fold belt in the north (Figure 1a,b). With Huaying Mountain and Longquan Mountain anticlinal belts as the boundary of the Sichuan Basin, it is divided into three tectonic zones, the southeastern tectonic area (including the high-steep fold belt in the eastern Sichuan Basin and the low-steep fold belt in the southern Sichuan Basin), the northwestern tectonic area (including the low-flat fold belt in the northern Sichuan Basin and the low-steep fold belt in the western Sichuan Basin), and the central Sichuan tectonic area between Huaying Mountain and Longquan Mountain (Figure 1b) [32–34]. Organic-rich shales are deposited mainly in Yibin-Luzhou city in the southern Sichuan Basin, Fuling-Shizhu city in the southeastern margin and Lichuan city in western Hubei Province in the eastern margin. The organic-rich shales are the thickest in the Changning-Fushun and Fuling-Shizhu area. At present, the shale gas demonstration area and high-yield shale gas wells in the Sichuan Basin are situated in the southern and southeastern areas [17].

The study stratum is the Longmaxi Formation of the Lower Silurian in the southern Sichuan Basin [35,36]. It is mainly a set of shallow–deep water shelf facies in the Longmaxi Formation, and stratum thickness is mainly 0–600 m. The lithology is mainly siliceous shale, silty shale and organic (carbonaceous) shale (Figure 1c). At the bottom of the Longmaxi Formation, there is a large amount of graptolite which gradually decreases upward [37].

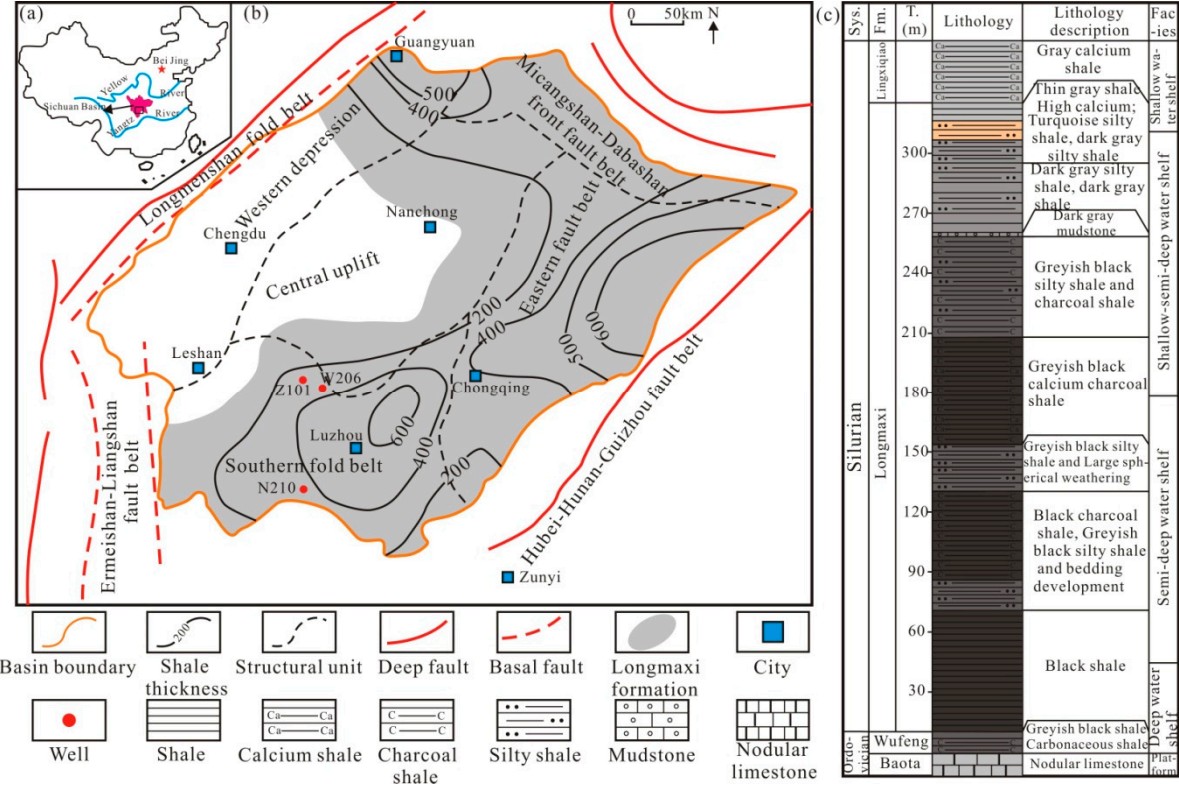

**Figure 1.** Location, tectonic setting and stratigraphy of the study area. (**a**) Geographic location of the Sichuan Basin; (**b**) geographic location of the study area; (**c**) generalized stratigraphic columns of the Longmaxi Formation.

## 3. Materials and Methods

### 3.1. Sample Collection

A total of 40 shale samples (numbered from N1-1 to N1-20 and N10-1 to N10-20) from the Silurian Longmaxi Formation were collected from three wells (Z101, W206 and N210) in the southern Sichuan Basin. TOC analysis of all the shale samples using a CS230HC carbon and sulphur analyser. The pore type and morphology of all the shale samples were observed by scanning electron microscopy (SEM), argon ion polishing technology and poly-ion beam scanning electron microscopy (FIB-SEM). A high-pressure mercury injection method was used to determine the pore size and distribution of mesopores and macropores (N1-2, N1-5, N1-9, N1-11, N1-13, N1-20, N10-2, N10-4, N10-5, N10-6, N10-8, N10-10, N10-12, N10-13, N10-15), and micropore distribution used the low-pressure nitrogen adsorption method (N1-8, N1-9, N1-11, N1-8, N1-9, N1-11,N10-1, N10-2, N10-4, N10-5, N10-6, N10-7, N10-10, N10-13). Then, the fractal dimensions of different pore sizes were calculated [38–40]. All of the experiments were conducted at the State Key Laboratory of Oil and Gas Reservoir Geology and Exploitation, Southwest Petroleum University, China.

### 3.2. Low Pressure N₂ Adsorption Analysis

The important parameters of the shale pore mainly include adsorption and desorption isotherms, specific surface, specific pore volume and pore size distribution, which can be obtained by testing shale samples with the nitrogen adsorption experiment. The nitrogen adsorption isotherm refers to the nitrogen adsorption capacity varying with relative pressure at the constant temperature. According to the Chinese National Standard GB/T 5751-2009 [41], the pore size and shape can be determined by the adsorption and desorption isotherms. $N_2$ adsorption analysis was conducted using a Nove 2000e automatic nitrogen adsorption instrument from Quantachrome Company, Boynton Beach, FL, USA. The specific surface measurement range was larger than 0.01 m2/g, the pore diameter range was 0.35–200 nm, the pore volume was larger than 0.0001 cc/g, and the pressure range was 0–0.13 MPa. First, the sample was made into particles approximately 3 mm, which were baked at 200 °C to remove impurity gas prior to the experiment. After natural cooling, 5 g of sample was taken out, and the adsorption–desorption isotherm curve of the sample was obtained by using nitrogen as the adsorption gas.

### 3.3. MIP Analysis

According to the Chinese Oil and Gas Industry Standard SY/T 5346-2005 [42], the pore distribution of mesopores and macropores was determined by mercury injection. The instrument used for these measurements was PoreMaster 60 mercury porosimeter produced by Quantachrome Company, USA, and its measurement range of the pore diameter was 3.6 nm–950 μm. The high-pressure mercury intrusion method can make up for the deficiency of the low-pressure nitrogen adsorption method in characterizing the macropores, and the samples used in the test were all the same.

### 3.4. Fractal Dimension Analysis

Fractal dimension can quantitatively reflect the distribution of pore size, mainly including the image analysis, the gas adsorption method and the mercury injection method [43–45]. As the fractal dimension obtained by image analysis is greatly influenced by the accuracy of the image processing and human factors, this paper mainly used the gas adsorption method and the mercury injection method. The distribution of pore diameter obtained by the two methods were superimposed, and the relationship between the pore number measurement N(d) and the pore diameter (d) was used to determine whether it met the fractal statistic in the full pore size range. Then, the fractal dimension in the different pore size was obtained.

In this paper, the fractal characteristics of micropore and mesopore were analysed by the nitrogen adsorption method [46], and the fractal dimension was calculated by the Frenkel–Halsey–Hill (FHH) model. The calculation method is as follows:

$$\ln V = (D_n - 3) \ln\left[\ln\left(\frac{P_0}{P}\right)\right] + C \tag{1}$$

$$Dn = \begin{cases} 3S + 3, \textit{Van Edward force} \\ S + 3, \textit{Desorption force} \end{cases} \tag{2}$$

where $V$ is the gas adsorption capacity at equilibrium pressure in cm³/g, $C$ is constant, $P_0$ is the gas saturated vapour pressure in MPa, $P$ is the gas equilibrium pressure in MPa, $D_n$ is the pore fractal dimension based on gas adsorption, and $S$ is the slope of linear fitting in double logarithmic coordinates. When $P/P_0 < 0.45$, it is mainly Van Edward force, and the capillary effect can be ignored. When $P/P_0 > 0.45$, it is mainly interfacial tension, and the capillary coagulation effect is prominent.

The fractal dimension of the macropore was evaluated by the mercury injection method. According to the functional relationship between the amount of mercury injection and the pressure, the pore size and the pore volume of different sizes could be determined. The fractal dimension of the pore

structure could be obtained from the variation characteristics of pore volume and pore size [47,48]. The calculation method is as follows:

$$\ln S_{Hg} = -(D_m - 2) \ln P_c + C \tag{3}$$

$$P_c = 4\sigma \cos \theta / \sigma \tag{4}$$

where $S_{Hg}$ is the cumulative volume of mercury flowing through the capillary with diameter (d) in mL/g, $\theta$ is the contact angle, $\sigma$ is the interfacial tension, $P_c$ is the capillary pressure in MPa, $C$ is constant, and $D_m$ is the macropore fractal dimension based on MIP.

## 4. Results

### 4.1. Micropore Types and Characteristics

According to the classification methods of scholars both domestic and abroad, the shale micropore can be divided into inorganic pores and organic pores. Inorganic pores can be divided into intergranular pores and intragranular pores, and the organic pores can be divided into organic pores, intergranular pores between organic and mineral grain.

Intergranular pore refers to the pore between grains or grains in the matrix. It is an important pore space in the study area, which can be further divided into intergranular pores of rock grains and intergranular pores of mineral grains [49]. The clastic particles of the Longmaxi Formation are mainly quartz and a small amount of feldspar, and the pore spaces between the clastic particles are called intergranular pores (Figure 2a). Quartz and feldspar also form intergranular pores with carbonate minerals such as calcite, dolomite and pyrite (Figure 2b). Intergranular pores were the main pore type in the study area, most of the pore shapes were elliptical or irregular, and the pore size varied from nanometre to micron.

In addition, pyrite, carbonate and clay also form intergranular pores with the pyrite, and the clay mineral forms intergranular pores with the clay mineral and with the calcite [50], among which the former two are the most common. The main clay mineral was illite (74%), and its grains were mostly plate-like or lamellar. Microscopically, intergranular pores of plate-like or slit shape were commonly developed in interlamellar layers of the illite (Figure 2c). The direction of pore extension was usually the same as the bedding direction of the clay mineral. The width of the pores varied from nanometre to micron, and the narrow linear extension could reach several microns. Pyrite was mostly developed in the form of raspberry aggregates, in which the pore size of a single pyrite crystal was usually hundreds of nanometres. Its morphology was usually related to the contact mode between crystals, mostly triangular or irregular polygon (Figure 2d).

Intragranular pores are formed by dissolution of clastic particles or mineral crystals during the diagenesis of some chemically unstable minerals, generally including intragranular dissolution pores or internal pores of biofossils [6]. There were a few intragranular pores in the study area and were mainly composed of intragranular dissolved pores. Many corrosion pits were formed by the dissolution of mineral particles, the pore size varied from tens of nanometres to several micrometres, and the pore morphology was mostly triangular or irregular polygon (Figure 2e).

The organic pores were distributed mainly in kerogen [51], with poor connectivity and isolated distribution, and most of the organic pores were round, elliptical or irregular, with pore sizes ranging from several to hundreds of nanometres (Figure 2f). Organic pores were the most abundant pore type in shale, which plays an important role in shale gas generation and reservoir.

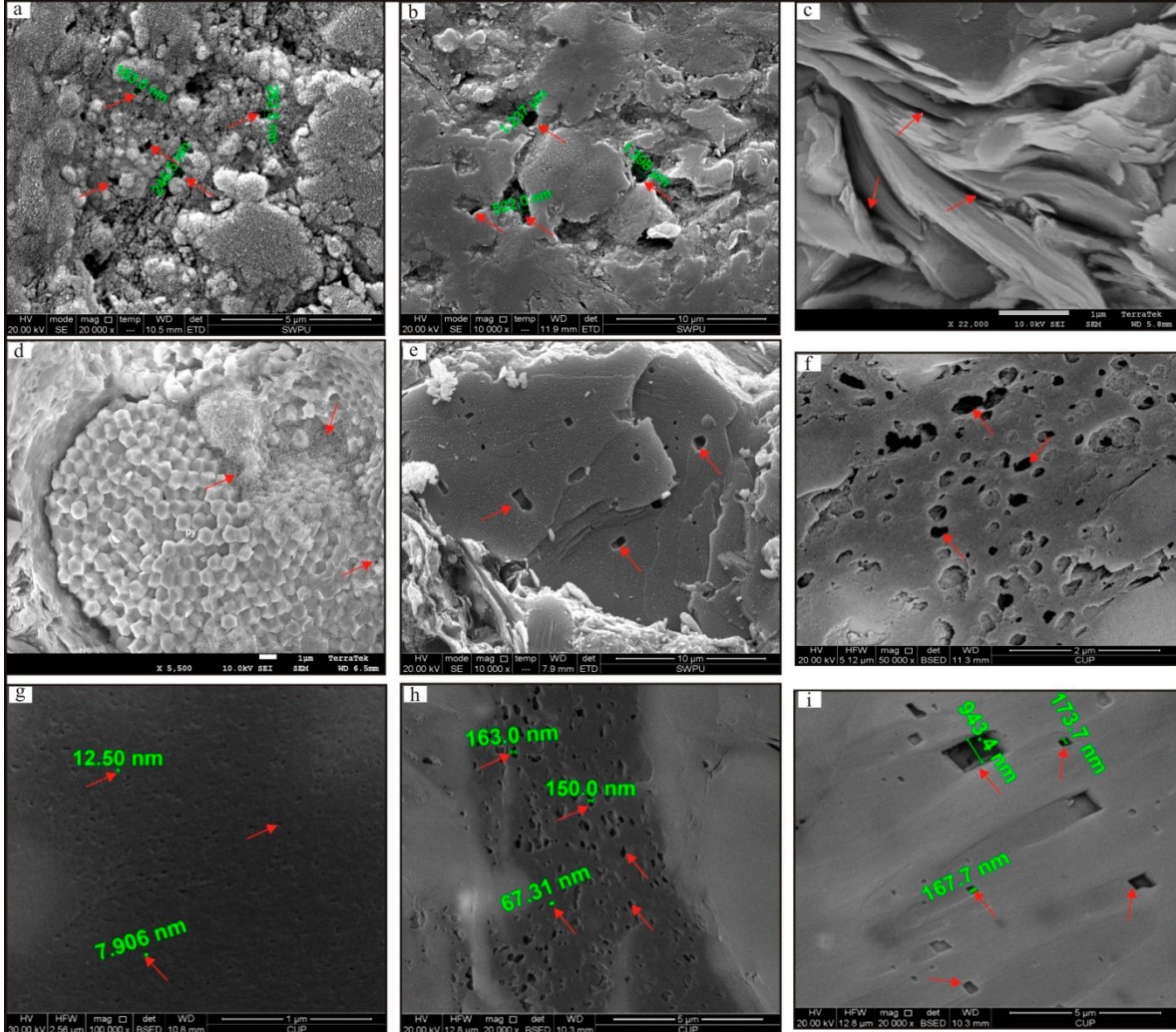

**Figure 2.** Microscopic pore characteristics of the Longmaxi shale in the southern Sichuan Basin. (**a**–**f**) SEM, (**g**–**i**) FIB-SEM. (**a**) Intergranular clastic pores, N210, 2208.80 m; (**b**) intergranular clastic porosity, N210, 2235.90 m; (**c**) clay inter-crystalline pore, Z101, 3388.80 m; (**d**) inter-crystalline pore, berry-shaped pyrite, Z101, 3457.20 m; (**e**) intragranular pore, corrosion pit, N210, 2224 m; (**f**) organic pore, N210, 2205.38 m; (**g**) nanoscale organic pore, W206, 3288.45 m; (**h**) nanoscale organic pores, W206, 3306.75 m; (**i**) intragranular pore, N210, 2251.69 m.

## 4.2. Pore Size Distribution Characteristics

Based on the International Union of Theoretical and Applied Chemistry (IUPAC) classification criteria for pore diameter [52], shale micropores can be divided into micropore (<2 nm), mesopore (2–50 nm) and macropore (>50 nm). This paper adopts this pore size classification.

### 4.2.1. Pore Size Characterization through Image Analysis

The shale samples were processed by argon ion polishing technology, and the pore structure was observed by FIB-SEM to characterize the pore size distribution quantitatively [53].

In this study, pore diameters of 10 nm–5 μm were counted (Figure 3). The pore size and distribution of various shale types were obviously different. Most of the pore size and distribution were related to the microstructure of the primary mineral or organic matter [54]. Organic pores were the main storage space for adsorbed shale gas, and most of the pore shape was honeycomb and they had good connectivity. The organic pore was mainly distributed in the range of 0–20 nm, followed by >100 nm, which was observed in the same organic matter particle or band, was well sorted and the heterogeneity

was weak (Figure 2g,h). The intergranular pore was the most widely distributed and well-connected in the electron microscope. The interlayer pore of the clay aggregate provided the adsorption site for shale gas. The pore size distribution was 0–10 nm and >100 nm, and the heterogeneity was strong (Figure 2c). The intragranular pore was the smallest in the study area, with poor connectivity, the largest variation range of pore size, the strongest heterogeneity and the smallest contribution to the shale reservoir (Figure 2i).

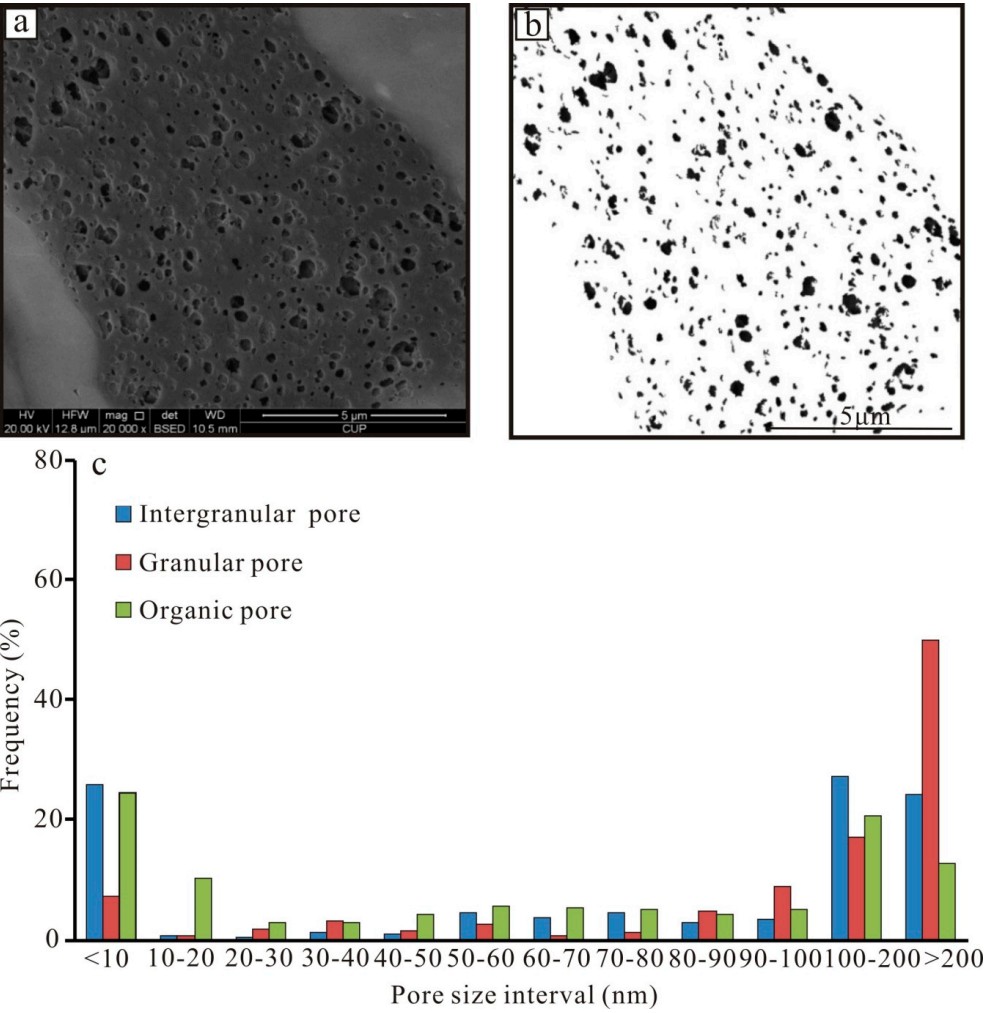

**Figure 3.** Pore size characterization through graphic analysis of the Longmaxi Formation in the southern Sichuan Basin. (**a**,**b**) Image processing schematic; (**c**) pore size distribution of different pore types.

### 4.2.2. Pore Size Characterization through Nitrogen Adsorption and Mercury Pressure Method

The optimum pore diameter of the low-pressure nitrogen adsorption method is 1–50 nm, it can hardly characterize macropores. The actual pore diameter of the high-pressure mercury injection method is more than 10 nm, but it is easy to cause artificial fractures and the stress sensitivity, resulting in error of testing results. Therefore, the pore size distribution can be characterized by the combination of the low-pressure nitrogen adsorption method and the high-pressure mercury injection method, which can better make up for the limitations of the two methods [55]. The distribution of micropore and mesopore in shale (pore diameter < 50 nm) was determined by low pressure nitrogen adsorption. However, the large pore size (>50 nm) was determined by the high-pressure mercury injection method, which could not only cover the overall pore size but also compensate for the testing limitations of the two methods.

The N$_2$ adsorption analysis was primarily employed to obtain the distribution characteristics of the shale mesopores. The shale pore volume was obtained by calculating the amount of liquid nitrogen occupying the pores based on the capillary condensation mechanism. Under a specific relative pressure, the capillary condensation phenomenon would occur in a pore with a specific diameter corresponding to the specific relative pressure. The volume of pores with different pore diameters can be obtained by calculating the amount of liquid nitrogen occupying the pores at different relative pressures according to the Barrette−Joynere−Halenda (BJH) method [56,57]. Because of the different mechanisms of nitrogen adsorption and desorption, the adsorption and desorption curves did not coincide in higher relative pressure ($P/P_0 > 0.4$), and the desorption curve was located at the top of the adsorption curve to form the hysteresis loop. The shapes of the capillary pressure curves are relatively consistent based on the analysis of the mercury injection curves [58]. The adsorption curves of the samples were slightly convex in the relatively low-pressure region, and the adsorption capacity increased rapidly. It was a transition from monolayer to multilayer, reflecting the micropore in the samples. In the middle relative pressure region, the adsorption capacity increased steadily, which was the adsorption process of the multilayer. In the relatively high-pressure region, the isotherm rose rapidly, and the curve rose near the saturated vapour pressure, which indicated that there were some macropores in the sample, and the desorption curve did not coincide with the adsorption curve to form the hysteresis loop (Figure 4). The shape of the desorption and adsorption curves in the study area was close to the type IV of the adsorption isotherm of the International Union of Theoretical and Applied Chemistry (IUPAC) (Figure 5) [59]. However, when the relative pressure was close to 1, there was no obvious platform segment, which had the characteristics of type I and II showing that there were mainly micropores, mesopores and a small number of macropores.

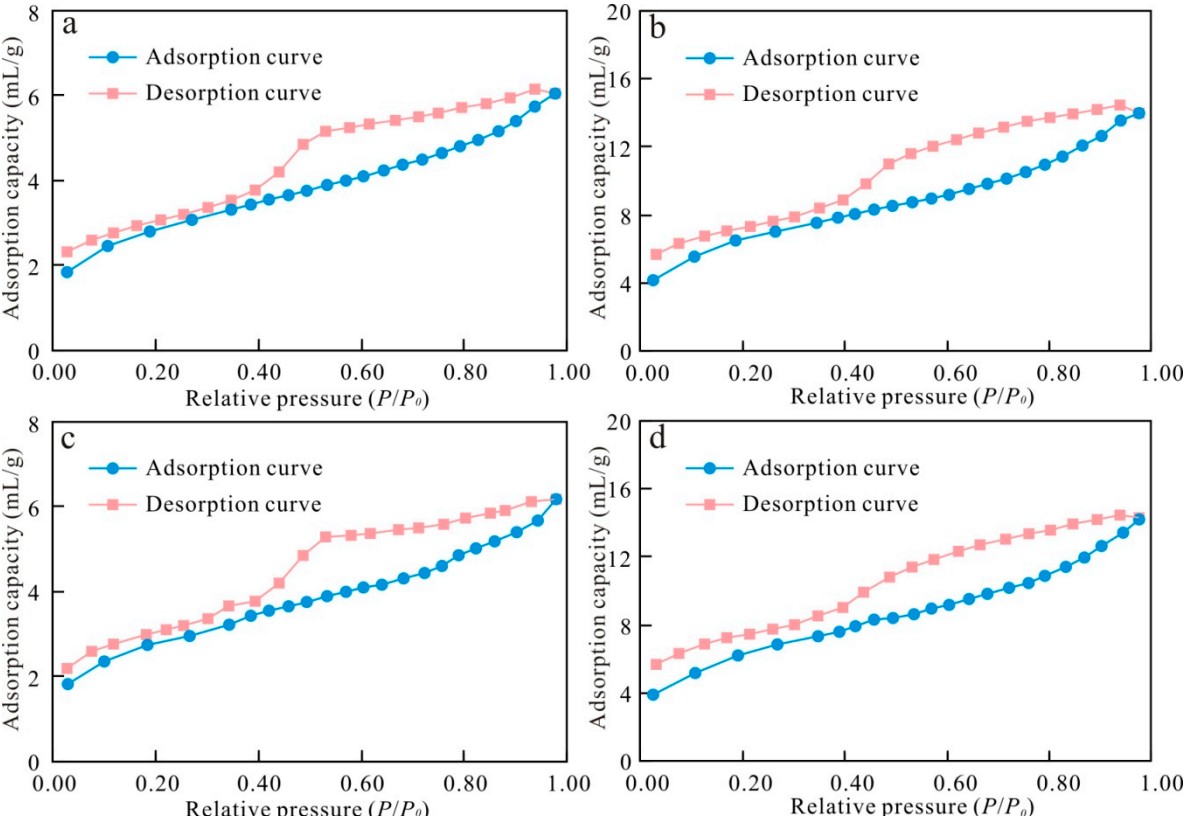

**Figure 4.** Isothermal curves of nitrogen adsorption and desorption of the Longmaxi Formation in the southern Sichuan Basin. (**a**) N1-1; (**b**) N1-2; (**c**) N10-17; (**d**) N10-18.

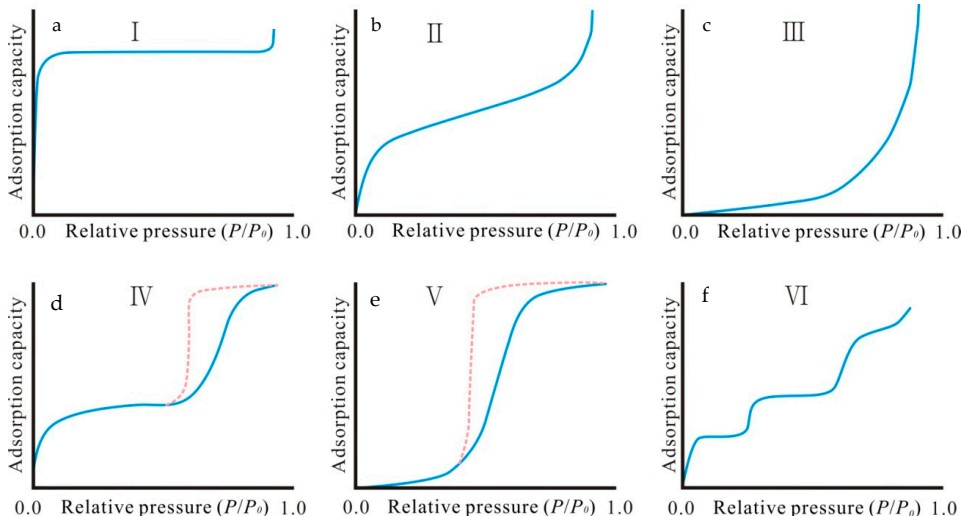

**Figure 5.** Types of adsorption isotherms by International Union of Theoretical and Applied Chemistry (IUPAC). (**a**) Type I; (**b**) Type II; (**c**) Type III; (**d**) Type IV; (**e**) Type V; (**f**) Type VI.

The BJH method was used to calculate the pore size distribution of shale samples in the study area (Table 1). The average pore diameter was 3.50–3.85 nm, the specific surface and pore volume were larger, which was conducive to the adsorption of shale gas. The specific surface was positively correlated with the pore volume. With the increase of the specific surface, the pore volume increased. The pore size distribution showed a single peak pattern, and the peak value was mainly 2–5 nm, which reflected the development of nanoscale pores in the samples. Mesopores accounted for 87.7% of the total pore volume, micropores accounted for 12.3%, and macropores were not detected (Figure 6).

**Table 1.** Pore size distribution of nitrogen adsorption of the Longmaxi Formation in the southern Sichuan Basin.

| No. | Specific Surface ($m^2$/g) | Specific Pore Volume (mL/g) | Average Aperture (nm) | Pore Volume (mL/g) | Micropore Proportion (%) | Mesopore Proportion (%) | Macropore Proportion (%) |
|---|---|---|---|---|---|---|---|
| N10-1 | 25.08 | 0.029 | 3.79 | 0.1048 | 12.47 | 87.53 | 0 |
| N10-5 | 24.01 | 0.019 | 3.50 | 0.1007 | 9.84 | 90.16 | 0 |
| N10-8 | 12.75 | 0.012 | 3.53 | 0.0641 | 16.09 | 83.91 | 0 |
| N10-17 | 23.13 | 0.020 | 3.60 | 0.0977 | 8.90 | 91.10 | 0 |
| N1-1 | 9.61 | 0.001 | 3.85 | 0.0455 | 14.25 | 85.75 | 0 |

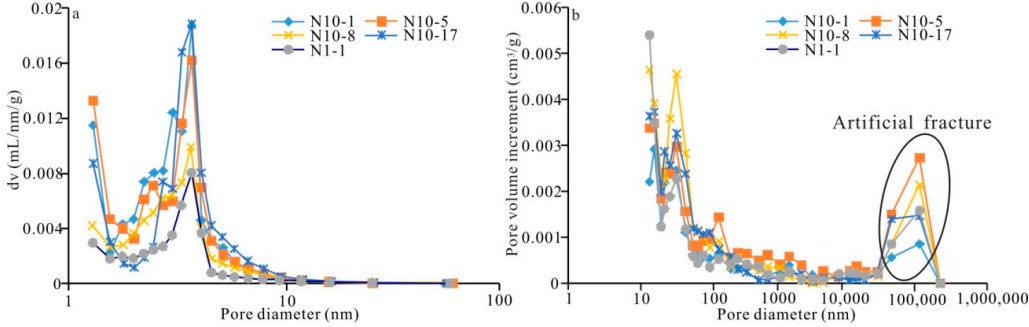

**Figure 6.** Pore size distribution curves of the shale samples of the Longmaxi Formation in the southern Sichuan Basin. (**a**) Low-pressure nitrogen adsorption method; (**b**) high-pressure mercury intrusion method.

The mercury volume corresponding to different pore sizes could represent the pore volume, and the distribution of macropore could be better measured by the high-pressure mercury intrusion method. Shale samples in the study area were mainly mesopores, accounting for 54.44–67.68% of the total pore volume. The pore size range of the macropores was wide (Table 2). The pore size distribution of high-pressure mercury injection were basically three or more peaks (Figure 7). They were approximately 30 nm, 100–1000 nm and 100 μm, showing that the peak value at 100 μm may be due to artificial fractures caused by excessive pressure during sample preparation or mercury injection. Therefore, this paper only included pore diameters less than 1 μm.

**Table 2.** Pore size distribution characteristics through the mercury pressure method of the Longmaxi Formation in the southern Sichuan Basin.

| No. | Micropore Proportion (%) | Mesopore Proportion (%) | Macropore Proportion (%) | | | Pore Volume (mL/g) |
|---|---|---|---|---|---|---|
| | | | 50–100 nm | 100–1000 nm | >1000 nm | |
| N10-1 | 0 | 65.71 | 8.24 | 12.21 | 13.83 | 0.022 |
| N10-5 | 0 | 54.44 | 7.68 | 16.11 | 21.76 | 0.033 |
| N10-8 | 0 | 67.68 | 9.46 | 10.20 | 12.66 | 0.036 |
| N10-17 | 0 | 65.53 | 10.88 | 10.06 | 13.53 | 0.031 |
| N1-1 | 0 | 66.57 | 6.40 | 11.00 | 16.03 | 0.025 |

Therefore, the curve of pore volume increment (△V) measured by the low-pressure nitrogen adsorption and the high-pressure mercury injection method with respect to pore diameter was characterized at 50 nm (Figure 7). Shale samples in the study area were mainly mesoporous, accounting for 25.43–56.01% of the total pores, followed by macropores, which have a wide pore size distribution, and micropores account for 11.33–17.73% (Table 3). The shale pore diameter was mainly in the range of 1 nm to 300 μm. There were three peaks, which were approximately 3 nm, 100 nm and 100 μm. The peak value of 10 μm to 300 μm may have been artificial fractures caused by excessive pressure in sample preparation or experiment [60–64].

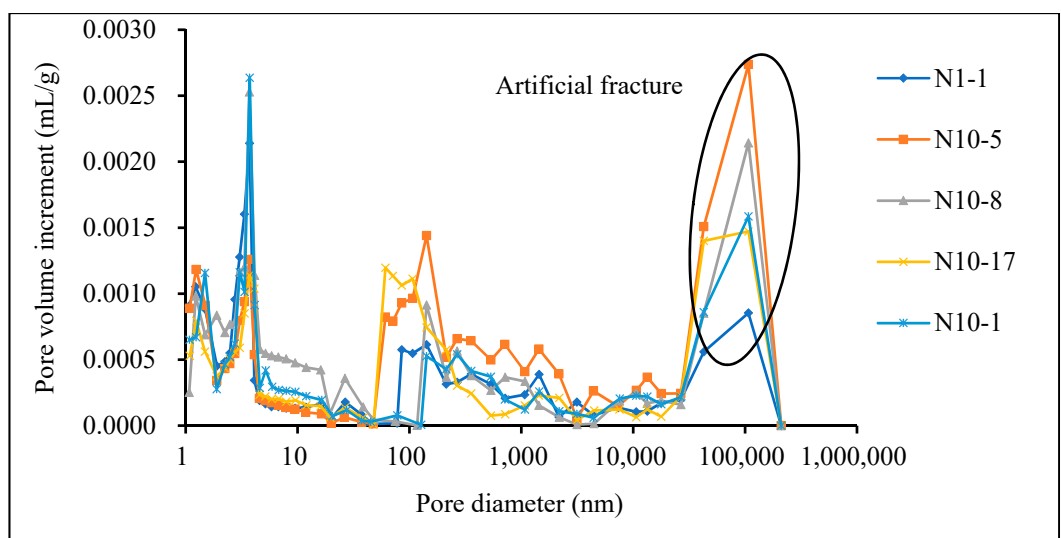

**Figure 7.** Pore size distribution characterized by nitrogen adsorption and high-pressure mercury intrusion of the Longmaxi Formation in the southern Sichuan Basin.

**Table 3.** Combined characterization of pore size distribution by nitrogen adsorption and high-pressure mercury intrusion of the Longmaxi Formation in the southern Sichuan Basin.

| No. | Micropore Proportion (%) | Mesopore Proportion (%) | Macropore Proportion (%) | | |
|---|---|---|---|---|---|
| | | | 50–100 nm | 100–1000 nm | >1000 nm |
| N10-1 | 17.73 | 47.78 | 3.20 | 14.63 | 16.56 |
| N10-5 | 13.46 | 25.43 | 10.30 | 21.62 | 29.19 |
| N10-8 | 11.93 | 56.01 | 0.14 | 12.31 | 19.56 |
| N10-17 | 11.33 | 34.69 | 17.04 | 15.76 | 21.19 |
| N1-1 | 14.32 | 42.13 | 5.93 | 16.13 | 23.67 |
| Average | 13.75 | 41.21 | 7.32 | 16.09 | 22.03 |

*4.3. Fractal Characteristics of Pore Structure*

4.3.1. Fractal Dimension of Micropores and Mesopores

According to the FHH model of nitrogen adsorption, the fitting curves of the shale samples showed obvious bilinear characteristics. At the stage of relatively low pressure ($P/P_0 < 0.45$), mainly Van Edward force, the capillary effect can be ignored. At the stage of relatively high pressure ($P/P_0 > 0.45$), mainly interfacial tension operates and the capillary coagulation effect is prominent [65]. In this paper, the curves of 11 shale samples were fitted (Figure 8, Table 4), and the overall correlation was good. In the Van Edward force stage, the fractal dimension ($D_{n1}$) ranged from 2.294 to 2.558, with an average of 2.469. In the capillary force stage, the fractal dimension ($D_{n2}$) ranged from 2.800 to 2.867, with an average of 2.838 (Table 4).

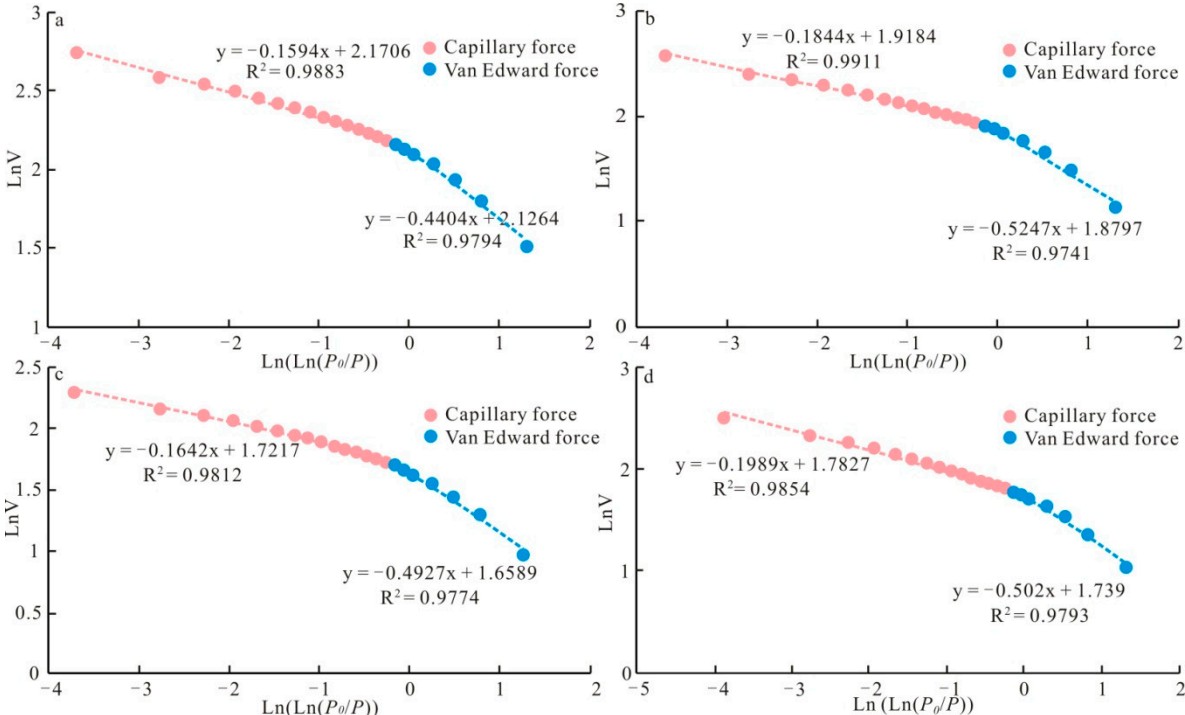

**Figure 8.** The fractal dimension calculated by the nitrogen adsorption method (partial samples) of the Longmaxi Formation in the southern Sichuan Basin. (**a**) N10-1; (**b**) N10-6; (**c**) N1-9; (**d**) N1-11.

**Table 4.** Fractal dimension calculated by the nitrogen adsorption method for the Longmaxi Formation in the southern Sichuan Basin.

| No. | $P/P_0 < 0.45$ | | | $P/P_0 > 0.45$ | | |
| | Fitting Equation | Fractal Dimension | Adj. R-Squared ($R^2$) | Fitting Equation | Fractal Dimension | Adj. R-Squared ($R^2$) |
|---|---|---|---|---|---|---|
| N10-1 | y = −0.440x + 2.126 | 2.569 | 0.979 | y = −0.159x + 2.171 | 2.841 | 0.988 |
| N10-2 | y = −0.433x + 1.838 | 2.567 | 0.987 | y = −0.169x + 1.887 | 2.831 | 0.983 |
| N10-4 | y = −0.538x + 1.712 | 2.463 | 0.958 | y = −0.133x + 1.792 | 2.867 | 0.963 |
| N10-5 | y = −0.489x + 2.079 | 2.511 | 0.973 | y = −1483x + 2.316 | 2.852 | 0.976 |
| N10-6 | y = −0.525x + 1.880 | 2.475 | 0.974 | y = −0.184x + 1.918 | 2.826 | 0.991 |
| N10-7 | y = −0.706x + 1.550 | 2.294 | 0.971 | y = −0.160x + 1.665 | 2.840 | 0.950 |
| N10-10 | y = −0.607x + 1.456 | 2.393 | 0.955 | y = −0.151x + 1.534 | 2.849 | 0.969 |
| N1-8 | y = −0.677x + 1.500 | 2.323 | 0.962 | y = −0.178x + 1.625 | 2.822 | 0.972 |
| N1-9 | y = −0.493x + 1.660 | 2.507 | 0.977 | y = −0.164x + 1.722 | 2.836 | 0.981 |
| N1-11 | y = −0.502x + 1.939 | 2.498 | 0.979 | y = −0.200x + 1.783 | 2.800 | 0.985 |
| N10-13 | y = −0.442x + 2.067 | 2.558 | 0.980 | y = −0.153x + 2.115 | 2.857 | 0.985 |

### 4.3.2. Fractal Dimension of Macropores

The mercury injection method is commonly used for evaluating pore structure. According to the functional relationship between the amount of mercury injection and the capillary pressure, the pore size and volume can be determined. The fractal dimension of pore structure can be obtained from the variation characteristics of pore volume and pore size [66,67]. In the mercury injection experiments, the difficulty of mercury entering the pore with a different pore size can be reflected by the capillary pressure. Therefore, the fractal dimension of the shale pore can be calculated based on the mercury injection data. According to the research results of other scholars [4,10], the pore size distribution through the mercury injection method is within the range of 10 nm–300 μm, so the fractal dimension mainly reflects the heterogeneity of the macropores [68]. Double logarithmic fitting of the mercury inflow curve with pore diameters greater than 50 nm was carried out (Figure 9). Except for one sample (N10-13), the fractal dimension of the other samples ranges from 2.534 to 2.924, with an average of 2.772 (Table 5), indicating that the macropore was highly heterogeneous.

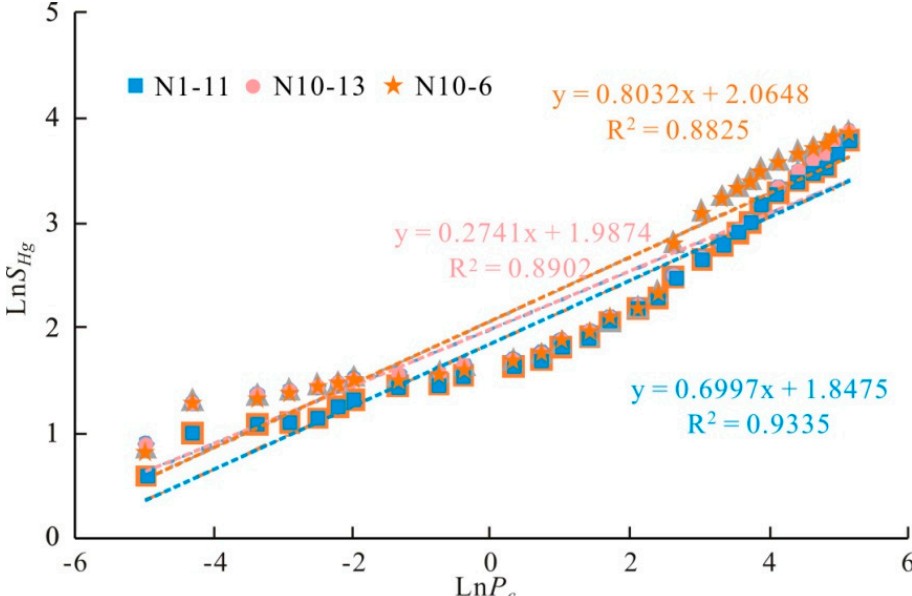

**Figure 9.** Pore fractal dimension through the mercury intrusion method for the Longmaxi Formation in the southern Sichuan Basin.

**Table 5.** Fractal dimension through the mercury intrusion method for the Longmaxi Formation in the southern Sichuan Basin.

| No. | Fitting Equation | Fractal Dimension | Adj. R-Squared ($R^2$) |
|---|---|---|---|
| N10-2 | y = 0.633x + 2.356 | 2.633 | 0.931 |
| N10-4 | y = 0.844x + 1.986 | 2.844 | 0.902 |
| N10-5 | y = 0.534x + 1.889 | 2.534 | 0.833 |
| N10-6 | y = 0.803x + 2.065 | 2.803 | 0.882 |
| N10-8 | y = 0.648x + 2.763 | 2.854 | 0.925 |
| N10-10 | y = 0.689x + 1.879 | 2.912 | 0.933 |
| N10-12 | y = 0.769x + 2.451 | 2.924 | 0.912 |
| N10-13 | y = 0.274x + 1.987 | 2.274 | 0.890 |
| N10-15 | y = 0.489x + 2.721 | 2.778 | 0.895 |
| N1-2 | y = 0.571x + 2.146 | 2.758 | 0.886 |
| N1-5 | y = 0.787x + 1.848 | 2.864 | 0.942 |
| N1-9 | y = 0.596x + 2.236 | 2.596 | 0.923 |
| N1-11 | y = 0.700x + 1.848 | 2.700 | 0.934 |
| N1-13 | y = 0.844x + 2.007 | 2.699 | 0.936 |
| N1-20 | y = 0.498x + 1.791 | 2.905 | 0.899 |

The pore fractal dimension was characterized by nitrogen adsorption and the high-pressure mercury intrusion method (Table 6); different methods reflect different size ranges. The fractal dimension of the micropores (1–4.5 nm) was relatively small, but its distribution was relatively scattered, which showed that the heterogeneity was relatively low. The fractal dimension of the mesopores (4.5–50 nm) was the largest, with small variation range, which indicated that the distribution of the mesopores has good self-similarity. There were similar characteristics in the fractal dimension of the macropores (larger than 50 nm), the fractal dimension was mainly 2.534–2.924, which indicated that the heterogeneity of the macropore was strong.

**Table 6.** Fractal dimensions of different pore sizes for the Longmaxi Formation in the southern Sichuan Basin.

| Calculation Method | Nitrogen Adsorption Method ($D_n$) | | Mercury Intrusion Method ($Dm$) |
|---|---|---|---|
| | In the van Edward Force Stage ($D_{n1}$) | In the Capillary Force Stage ($D_{n2}$) | |
| Diameter interval | 1–4.5 nm | 4.5–50 nm | >50 nm |
| Maximum value | 2.569 | 2.857 | 2.924 |
| Minimum value | 2.323 | 2.800 | 2.534 (removed N10-13) |
| Average value | 2.469 | 2.839 | 2.772 |

## 5. Discussion

### 5.1. Correlation of Pore Structure with Material Composition and TOC

Material composition plays an important role in formation and development of shale pores. Below we have divided the main minerals into three units (clay, silica + pyrite and feldspar + phosphate) [69] and discussed the relationship of pore structure with material composition and TOC.

In the study area, quartz content had a good positive correlation with specific surface and total pore volume (Figure 10a,b). Quartz had strong compaction resistance, the higher the quartz content was, the smaller the damage to the original pore was, and the more of the preserved pore was. For feldspar + carbonate, the clay content was negatively correlated with specific surface and total pore volume, with poor correlation (Figure 10c–f). During the diagenesis of shale, dissolution easily occurs [70]. The higher the content of feldspar + carbonate was, the more likely dissolution was, thus increasing the specific surface and total pore volume. However, the clay mineral had poor compaction resistance, and the pore volume associated with clay mineral was vulnerable to compaction. The difference

relationship between specific surface, total pore volume and mineral composition was related to the source of the quartz and the organic matter content, so it showed different degrees of correlation.

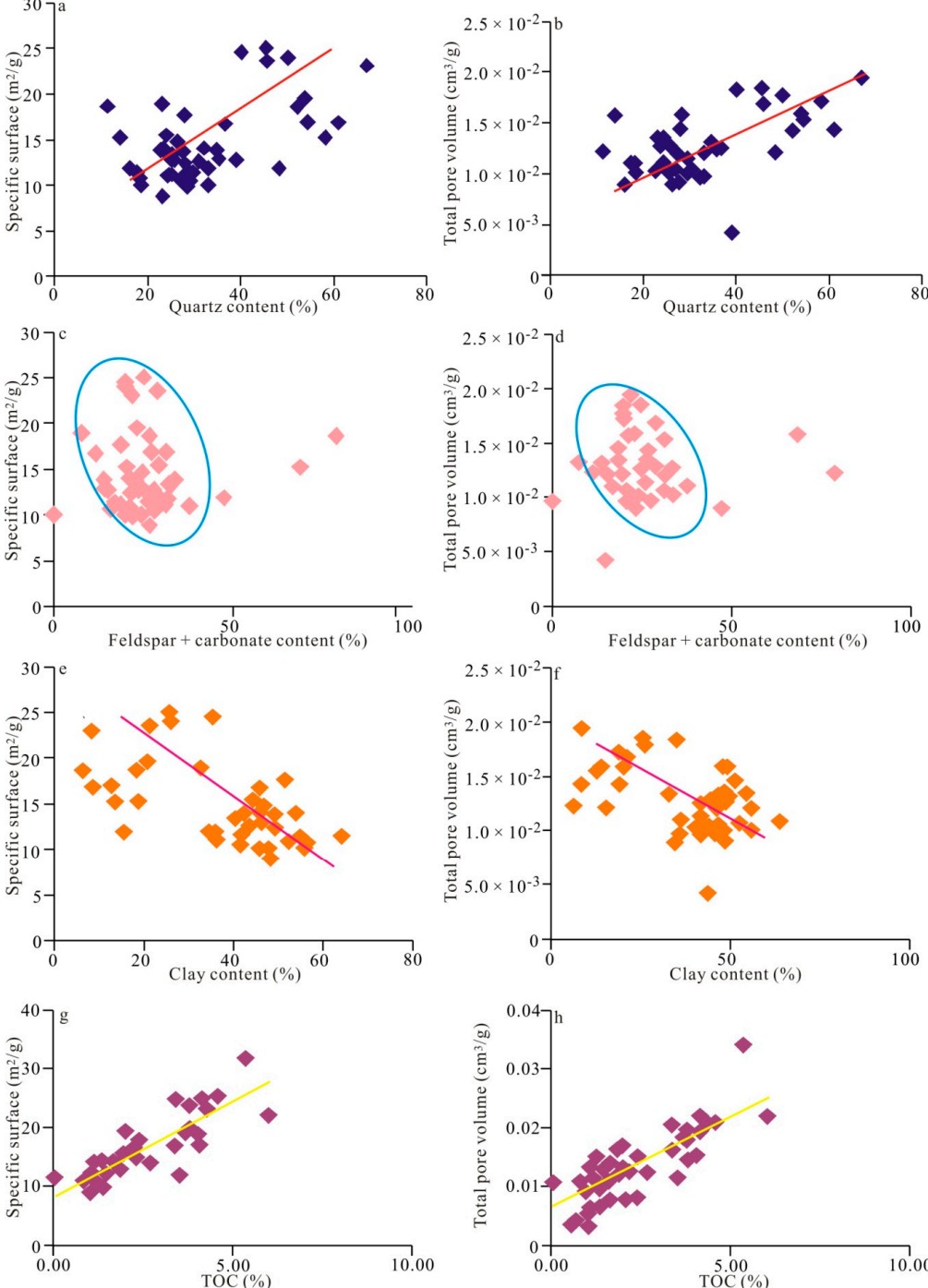

**Figure 10.** The correlation between material content and specific surface, pore volume for the Longmaxi Formation in the southern Sichuan Basin (n = 40). (**a**) The relationship between quartz content and specific surface; (**b**) The relationship between quartz content and total pore volume; (**c**) The relationship between feldspar + carbonate content and specific surface; (**d**) The relationship between feldspar + carbonate content and total pore volume; (**e**) The relationship between clay content and specific surface; (**f**) The relationship between clay content and total pore volume; (**g**) The relationship betweenTOC and specific surface; (**h**) The relationship betweenTOC and total pore volume.

Organic matter provides the material basis for organic pores in shale [71,72]. The type, maturity and abundance of organic matter have an important influence on formation and structure of organic pores. By analysing the relationship between TOC and specific surface, pore volume of shale samples, the positive correlation between them was well defined (Figure 10g,h). In addition, from the pore size distribution curves of the nitrogen adsorption of different TOC samples, the higher the TOC was, the larger the volume of micropores and mesopores (Figure 11). The higher the TOC was, the more favourable the development of micropores and mesopores, which was the key factor to control the pore volume.

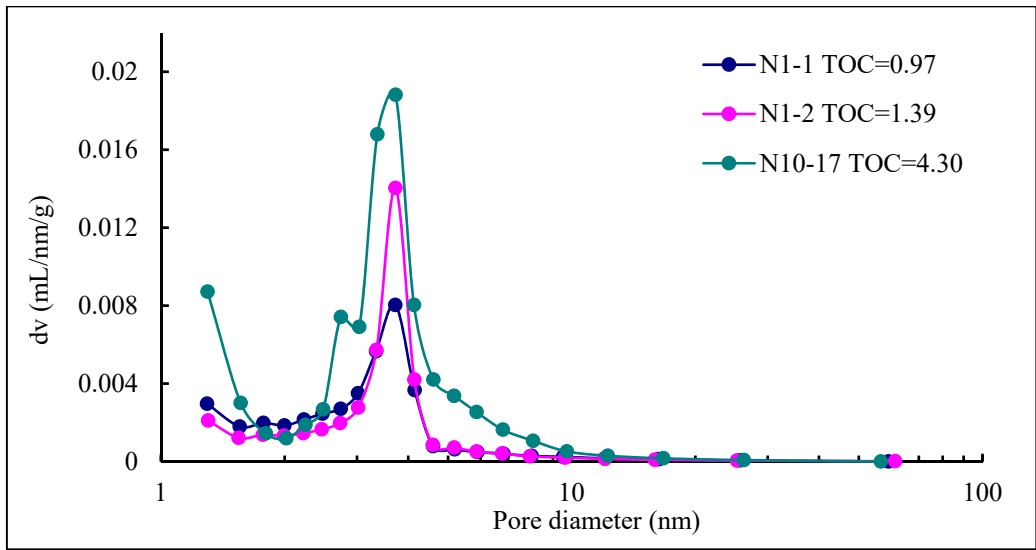

**Figure 11.** The relationship between total organic carbon (TOC), specific surface and pore volume (n = 40).

*5.2. The Correlation between Fractal Dimension with Mineral Composition and TOC*

The fractal dimension of shale pores was the result of material composition, thermal maturity, TOC and other factors. The correlation between the fractal dimension of different pore size and quartz and clay was quite different (Figure 12a–c). The correlation of the micropores was the best, followed by the mesopores, and the content of quartz and clay was the main controlling factor for the heterogeneity of the micropores. The influence of quartz and clay on the fractal dimension of the mesopores was relatively small. As the calcite in carbonate was mainly in the form of cement, the carbonate content was negatively correlated with the fractal dimension of the mesopores. The pores or fractures always filled with cements, the specific surface was reduced, which decreased the fractal dimension of the pore structure. The fractal dimension of the macropores had the worst correlation with the content of quartz and clay, but there was greater positive correlation between carbonate minerals and the heterogeneity of the macropore, which was related to the intragranular dissolved pore provided by the carbonate mineral. In this study, the TOC of shale samples was 1.19–4.59%, indicating high degree of thermal evolution [73–75]. Within this range, the fractal dimension of micropores and mesopores shows the weak correlation with TOC. However, with the increase of TOC, the fractal dimension of the macropores increased (Figure 12d).

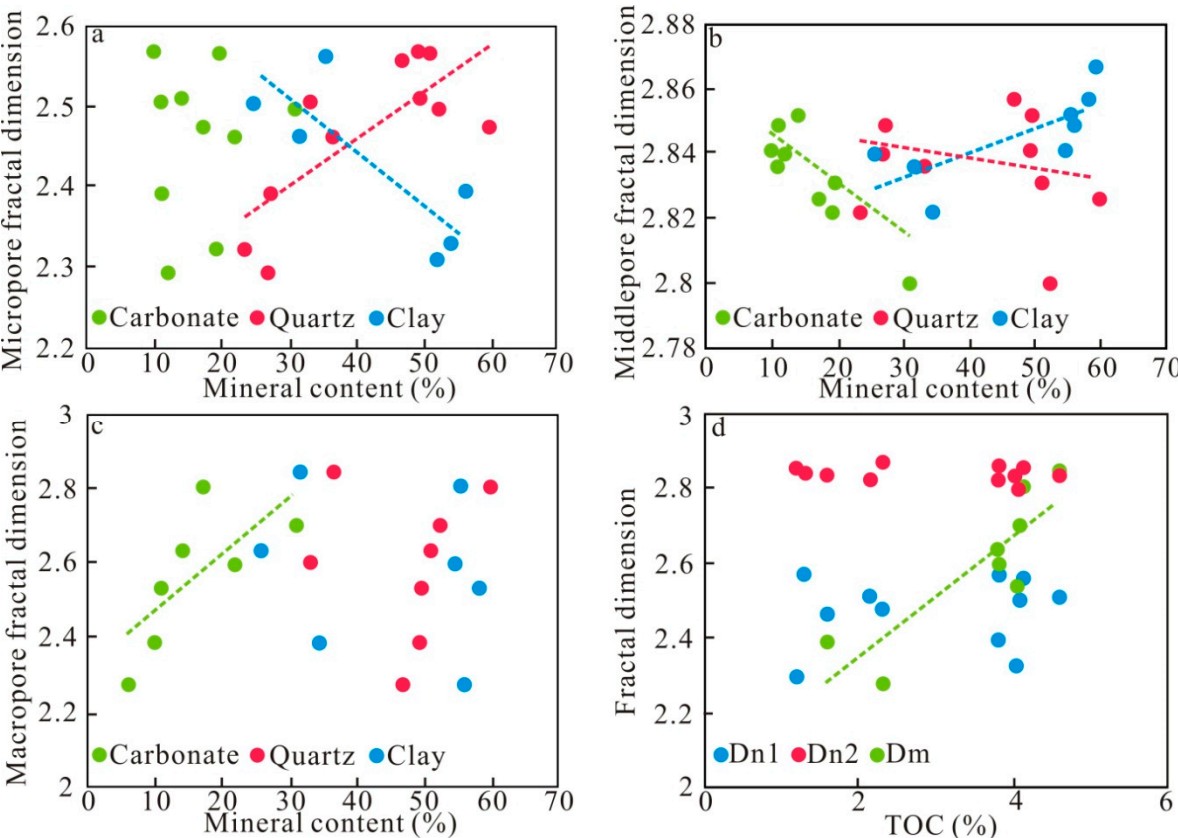

**Figure 12.** The relationship between fractal dimension with mineral composition and TOC. (**a**) Mineral content and fractal dimension of the micropore; (**b**) mineral content and fractal dimension of the micropore; (**c**) mineral content and fractal dimension of the macropore; (**d**) TOC and pore fractal dimension.

## 6. Conclusions

The heterogeneous pore structure in siliceous shale of the Silurian Longmaxi Formation were analysed using FE-SEM observations, low pressure $N_2$ adsorption, MIP and the fractal method. The conclusions drawn from this study were as follows:

(1) There were three types of pores: intragranular pore, intergranular pore and organic pore; organic pores were the main storage space for adsorbed shale gas. Based on the argon ion polishing + FIB-SEM, the pore size of organic pores was mainly 0–20 nm, and intergranular pores was 0–10 nm and >100 nm.

(2) A new method for evaluating heterogeneity of shale pores was established. The size distribution of micropore and mesopore in siliceous shale (<50 nm) was determined by low pressure nitrogen adsorption, the large pore size (>50 nm) was determined by the high-pressure mercury injection method, and the fractal dimensions of different pore sizes were calculated respectively. The combination of the two methods could characterize well the size distribution of the micropores.

(3) The mesopore of 2–20 nm was the main type, followed by the macropore of 100–300 nm. The average fractal dimension of the mesopores (4.5–50 nm) was 2.839, and the macropores (larger than 50 nm) was 2.772, indicating a strong heterogeneity and complex pore structure. The correlation between the fractal dimension of different pore sizes and quartz and clay of the Longmaxi Formation shale was quite different. Quartz content was positively correlated with specific surface and total pore volume.

**Author Contributions:** Conceptualization, investigation and writing—original draft preparation, H.L.; methodology, H.T.; software, M.Z.; validation of model, H.L.; writing—review and editing, H.L. and M.Z.; project administration and funding acquisition, H.T.; English polishing and editing, H.L. and M.Z.

**Funding:** This research was funded by the Natural Science Foundation of China, grant number 51674211 and the Key Projects of the Natural Science Foundation, China, grant number 51534006).

**Acknowledgments:** We are very grateful to Southwest Oil and Gas Field Company, PetroChina, for their kind help and support to complete this study and the permission to publish the results. We thank all the editors and reviewers for their helpful comments and suggestions.

**Conflicts of Interest:** The authors declare no conflict of interest. The funders had no role in the design of the study; in the collection, analyses, or interpretation of data; in the writing of the manuscript, or in the decision to publish the results.

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
