# Peer review of "Micropore Structural Heterogeneity of Siliceous Shale Reservoir of the Longmaxi Formation in the Southern Sichuan Basin, China"

_minerals, doi:10.3390/min9090548_

Round 1
Reviewer 1 Report
The marine shale of the southeastern Sichuan Basin in China possesses a highly mature and strongly heterogeneous pore structure, and studying complex pore structures is the key to understanding the mechanism of shale gas accumulation. The article can be accepted for publication. However, there are still some issues in the article, and the authors need to revise them carefully:
1. There are many long sentences throughout the manuscript. Long sentences harm the clarity. The authors should break these sentences or use 'and' for connections. Also, there are some grammatical errors.
2. How do you determine that the pores of SEM are true pores, not the artificial pores produced during the manufacturing process? Should the pictures add the stratum, Wufeng Formation or Longmaxi Formation?
3. Artificial fractures should be added in Figure 6.
4. Refer to the published articles in “Minerals”, review the conclusions and summaries.
5. There is a lack of relevant references. In recent years, there have been many articles about shale gas published in international journals, such as “Fuel”, “Minerals”, “Marine and Petroleum Geology”, “Journal of Natural Gas Science and Engineering” etc.
Author Response
Answers to reviewers:
The inappropriateness of the English sentences in the article has been modified, including some grammatical errors, professional vocabulary, etc. The artificial pores that produced during the manufacturing process are micron scale, but the pore size of SEM ranges from tens to hundreds of nanometers. The study stratum is the Longmaxi Formation of the Lower Silurian in the southern Sichuan Basin,we have deleted the Wufeng formation. We have added the artificial fractures in Figure 6 (Fig.7). Refer to the published articles in “Minerals”, we have rewritten the conclusions and summaries. We have added a lot of relevant references.Thank you for the kind advice.
Sincerely yours,
Hu Li Hongming Tang
Reviewer 2 Report
The paper discusses a relevant topic/data from a relevant area, but especially the discussion and conclusions need to be significantly improved and partly completely rewritten. English should certainly also be improved, including the technical vocabulary which shows signs of words incorrectly being used.

Author Response
Answers to reviewers:
We have substantially revised our manuscript after reading your comments. A revised manuscript with the correction sections red marked was attached as the supplemental material and for easy check/editing purpose.
Refer to the published articles in “Minerals”, we have rewritten the conclusions and summaries, especially conclusions. We have added the relevant references. Thank you for your opinion about rock type. Previous studies have confirmed that the shale of Longmaxi Formation in the study area has the characteristics of high Qz content and low clay content, high Qz content is related to secondary quartz, the grain size of quartz has reached the clay level. According to the grain size, characteristics and formation mechanism of rocks,scholars in China called these rocks as shale in the study area, which is the common understanding of this area. Silica content plays an important role in shale fracturability and “Sweet spots zone”evaluation. According to your opinion, we chang the type of shale in this paper to siliceous shale. According to your opinion, we have responded one by one in the annex. Should you have any questions, please contact us without hesitate.Thank very much for your kind advice.
Sincerely yours,
Hu Li Hongming Tang

Round 2
Reviewer 2 Report
Nine revision comments were not taken into account or inappropriately addressed. As an author, you are not allowed to ignore comments from reviewers. If you do no agree with comments, then address them in an accompanying letter.
In the attached document, the relevant remarks have been retaken. Also pay attention to the following general remark:
Use only following powers to cite number 10^3, 10^6, 10^9...

Author Response
After reading the nine revision comments, we try our best to substantially revised our manuscript and reply each question.
Meanwhile, the editors at AJE have edited for proper English language, grammar, punctuation, spelling, and overall style by one or more of the highly qualified native English speaking of the paper. This certificate may be verifed
on the AJE website using the verification code 4F07-D270-F4C7-BAAF-F23D.
General remark has also been revised.
We look forward to hearing from you.
Sincerely yours,
Hu Li, Hongming Tang, Majia Zheng
